# Evaluation of a Serum-Free Medium for Human Epithelial and Stromal Cell Culture

**DOI:** 10.3390/ijms231710035

**Published:** 2022-09-02

**Authors:** Christophe Caneparo, Stéphane Chabaud, Julie Fradette, Stéphane Bolduc

**Affiliations:** 1Centre de Recherche en Organogénèse Expérimentale de l’Université Laval/LOEX, Axe Médecine Régénératrice, Centre de Recherche du CHU de Québec-Université Laval, Quebec, QC G1J 1Z4, Canada; 2Centre de Recherche sur le Cancer, Université Laval, Quebec, QC G1R 3S3, Canada; 3Department of Surgery, Faculty of Medicine, Université Laval, Quebec, QC G1V 0A6, Canada

**Keywords:** serum-free medium, cell culture, serum-containing medium, cellular morphology, proliferation, clonogenicity, metabolism

## Abstract

Over the past decade, growing demand from many domains (research, cosmetics, pharmaceutical industries, etc.) has given rise to significant expansion of the number of in vitro cell cultures. Despite the widespread use of fetal bovine serum, many issues remain. Among them, the whole constitution of most serums remains unknown and is subject to significant variations. Furthermore, the presence of potential contamination and xenogeny elements is challenging for clinical applications, while limited production is an obstacle to the growing demand. To circumvent these issues, a Serum-Free Medium (SFM) has been developed to culture dermal and vesical fibroblasts and their corresponding epithelial cells, namely, keratinocytes and urothelial cells. To assess the impact of SFM on these cells, proliferation, clonogenic and metabolic assays have been compared over three passages to conditions associated with the use of a classic Fetal Bovine Serum-Containing Medium (FBSCM). The results showed that the SFM enabled fibroblast and epithelial cell proliferation while maintaining a morphology, cell size and metabolism similar to those of FBSCM. SFM has repeatedly been found to be better suited for epithelial cell proliferation and clonogenicity. Fibroblasts and epithelial cells also showed more significant mitochondrial metabolism in the SFM compared to the FBSCM condition. However, the SFM may need further optimization to improve fibroblast proliferation.

## 1. Introduction

Fetal bovine serum is the most commonly used supplement for cell culture. This and other serums allow cell culture in vitro (proliferation, differentiation, etc.). In contrast, culture media provide carrier proteins, essential nutrients, hormones and growth factors, but also promote cell attachment to the plastic surface. Due to the increasing use of in vitro cell cultures for applications in pharmaceutical, cosmetic or fundamental research, the demand for serums is growing by as much as 10–15% annually [1], causing a three-fold increase in the price of serums [2].

Despite the wide use of fetal bovine serum (FBS), many inconvenient aspects remain. The serum is composed of more than 1800 proteins and 4000 metabolites [3], and its whole composition remains largely unknown and is subject to significant variation between batches. Indeed, the serum is manufactured from pools of animals, contributing to the variability due to their genetic background and diet and the serum manufacturing process. Due to the responsiveness of the cells to their microenvironments, variation in serum composition can influence its fundamental biological properties [4], which poses a challenge for researchers in terms of obtaining consistent results [2]. To standardize serum batches, a screening procedure can be done to assess some of the constituents, such as the total proteins, immunoglobulins and growth factors, and to confirm its sterility, pH and osmolality [1]. Regulators, including the World Health Organization, thoroughly monitor serum production, which requires it to be traceable, originating from cattle that are free of bovine spongiform encephalopathy which have not ingested prohibited animal feeds [1]. Only one-third of the FBS produced in 2002 met the regulatory requirements for cell therapy applications [5,6], showing the difficulty of obtaining high-quality serums and reducing its potential availability for therapeutic assays. This is in contrast with the previously described increase in demand, which has been exacerbated by the fact that serum is a by-product of the meat industry. Indeed, it is not economically viable for livestock producers to increase herd sizes to produce FBS, further limiting the increase of the quantities of serum that will be available in the future. This suggests that global demand for serum may soon exceed supply [2,3]. Furthermore, despite meticulous analyses, the presence of potentially infectious agents in serums, particularly viruses or prions whose size is very small, remains possible, thereby making the filtration process inefficient [1]. Finally, ethical issues due to the potential suffering of bovine fetuses have been raised [5,6].

To circumvent these issues, alternatives to serums have been developed. Among them, human umbilical cord serum [7], human platelet lysate (HPL) [8,9], human serum [10] and chemically defined media [11] have received the most attention [1]. However, media with human-derived supplements share some of the disadvantages of fetal bovine serum-containing media (FBSCM) (e.g., variability between lots, manufacturing process) [12,13] and remain controversial because of a lack of availability and the possibility of disease transmission between donors. On the other hand, chemically defined media are, for the most part, xenogenic-free but also extremely expensive and generally only suitable for specific cell types.

In this context, developing a new serum-free medium (SFM) for cell cultures using different cell types (epithelial and stromal) would reduce variability and overall price while also providing higher security for translational and clinical studies, thus facilitating the translation from pre-clinical to clinical work compared to the use of serum. This study aims to evaluate the suitability of SFM for the culture of human primary epithelial and stromal cells, as they are more challenging to culture compared to immortalized or cancer cell lines. Thus, the performance of this SFM was assessed on populations of keratinocytes (KC) and urothelial cells (UC), as well as dermal fibroblast (DF) and vesical fibroblast (VF) populations. The hypothesis is that culturing these cells in SFM will bring about similar or improved cell proliferation rates compared to cultures using FBSCM over numerous passages. We report comparative results related to microscopic appearance, proliferation, cell size, clonogenicity and metabolism profiles for each cell type over three passages in these media.

## 2. Results

### 2.1. Microscopic Appearance Assessment

The morphology of three populations of keratinocytes (KC), urothelial cells (UC), dermal fibroblasts (DF) and vesical fibroblasts (VF) were evaluated microscopically (see Figure 1 and Table 1). A representative image of each cell type was selected for each cell population. The morphologies of the two other populations, which were from different donors, can be found in the Appendix A. No significant difference in cell morphology was observed for epithelial cells or fibroblasts during the three passages in the two media.

Theoretically, the proliferative potential of cells is inversely proportional to their size [14,15]. Thus, cell sizes were evaluated in each medium and on each day during the three passages. The results are represented as a curve to facilitate the interpretation. The results for each population are found in the Appendix A.

The KC and UC sizes varied between 15 to 17.5 μm in diameter, with no significant differences, as the curves roughly overlapped throughout the three passages (Figure 2). Concerning the DF and VF populations, cell size varied between 17 to 21 μm in diameter, with a slightly larger cell size for the latter in SFM compared to FBSCM. In comparison, no clear tendency was observed for the DF over the three passages.

### 2.2. Doubling Time Measurement

The KC and UC population doubling time was relatively higher at Passage 1 in FBSCM compared to SFM. It became roughly the same for the following passages (see Figure 3). The doubling times of the DF and VF in both media were similar for the first two passages of expansion. However, these lengthened in SFM condition on the third passage. For the epithelial cells, a high degree of variability was observed between the three populations (N = 3) in FBSCM compared to SFM. In contrast, for the fibroblast cells, the variability was higher in SFM compared to FBSCM as the culture progressed over time. However, such variability was not found within each population. Each of the three keratinocyte populations showed a statistically higher doubling time in FBSCM condition compared to SFM at Passage 1 (Appendix A).

### 2.3. Epithelial Clonogenicity Evaluation

Preservation of the stemness of epithelial cells is essential for their long-term use. Thus, clonogenicity was assessed using a coloration technique [16]. As counting and evaluating the colony-forming unit (CFU) can sometimes be challenging, two techniques were used and compared. The first was numbering, done by a relatively subjective observer due to the difficulty of separating overlapping colonies (Appendix A). Thus, to strengthen the evaluation and remove the observer bias, a second estimation used a Typhoon scanner^®^, which determined the stained surface area, i.e., the surface colonized by the epithelial cells.

The keratinocyte populations showed a globally higher, although not statistically significant, amount of CFU in SFM compared to FBSCM at Passage 1 (Figure 4A). This trend then decreased in subsequent passages. Similar results were found using the stained surface area (see Figure 4B). However, the CFU count/stain surface remained statistically higher (*p*-value < 0.0001) in SFM compared to FBSCM for Population 1 (Appendix A). Concerning the urothelial cells, more CFU were produced using SFM compared to FBSCM throughout the three passages (Figure 4C,D). Due to the high variability between the results of each population, this difference remained non-significant. However, the UC CFU count/stain surface was significantly greater in SFM conditions compared to FBSCM for all three populations and passages when data were not pooled together (Appendix A). Furthermore, keratinocytes cultured just after extraction for six days were statistically more numerous in SFM compared to FBSCM (Appendix A).

### 2.4. Metabolism Evaluation

Cellular metabolism was assessed using a Seahorse XFe96 extracellular flux analyzer. Energy metabolism values were normalized according to the number of cells using a CyQuant Cell proliferation assay kit.

No significant differences in keratinocyte basal and maximal extracellular acidification rate (ECAR) were observed concerning the media used (Figure 5). In contrast, a significantly higher maximal oxygen consumption rate (OCR) was observed when cells were cultured in SFM compared to FBSCM. For UC populations, a trend toward increased basal and maximal ECAR was observed with SFM compared to FBSCM. In the same way, the basal and maximal OCR were slightly and statistically increased in SFM compared to FBSCM.

On the other hand, DF showed a roughly lower but not statistically significant basal ECAR when cultured in SFM compared to FBSCM, while the maximal ECAR remained unchanged. In addition, the DF basal and maximal OCR were significantly higher when the cells were cultured in SFM compared to FBSCM. Finally, no significant differences were observed for the VF populations in basal or maximal ECAR. However, basal OCR was slightly higher when cells were cultured in SFM compared to FBSCM. Overall, an increase in the OCR was observed in SFM compared to FBSCM, regardless of cell type (Table 2).

### 2.5. SFM and FBSCM Cost Evaluation

Finally, the cost of the medium is important, as it can represent a large part of the expenses incurred by a laboratory. Therefore, SFM and FBSCM constituent prices were estimated using common purchasing platforms.

The total cost for a 500 mL bottle of SFM was lower than for FBSCM for fibroblasts by about 25 US dollars (one-third less) and roughly similar to FBSCM for epithelial cells (about 5 US dollars less) (Table 3). The three most expensive components of SFM are the insulin-transferrin-selenium, the L-glutamax and the lipid concentrate. In contrast, the FBS represents more than 99% of the cost for fibroblast FBSCM and 75% of FBSCM for epithelial cells.

## 3. Discussion

A SFM was formulated and assessed in this study to circumvent the various issues associated with FBS. The aim was to evaluate the feasibility of replacing FBSCM with a SFM with a relatively simple constitution but which nonetheless contained the essential elements for cell culture. Among them, essential amino acids, a carbon source, lipids, ion carriers, hormones and growth factors, and basic salts [1] were considered. The concentrations used in this SFM were primarily based on a physiological concentration to contrast with constitution of FBSCM. This safe and defined composition will also facilitate the clinical translation of various tissue-engineered substitutes produced following cell culturing. To assess the efficacy of the SFM, we evaluated the cellular characteristics of the primary KC, UC, DF and VF populations, i.e., the microscopic morphology, cell size, doubling time and glycolytic and mitochondrial metabolism. We also investigated the clonogenicity of the epithelial populations.

The morphology of epithelial cells and fibroblasts showed no notable difference, regardless of the medium in which they were cultured (Figure 1). This observation correlates with the measured similar cell sizes (Figure 2). In their study, Chase et al. observed a drastic change in cell morphology when cultivating mesenchymal stem cells in a SFM [17]. Indeed, the cells displayed a spindle-shaped morphology instead of a more flattened fibroblast-like one, indicating that some components may have been missing to allow regular cell maintenance. In contrast, Taihi et al. used HPL to cultivate mesenchymal stem cells. Similarly, they did not observe microscopic differences in cell morphology, regardless of the concentration of HPL used [18]. Therefore, the SFM presented in this study contained components which provided signals similar to those of FBSCM, thereby allowing cells to have their characteristic appearance when cultured in 2D. These results are exciting, as epithelial cells are challenging to cultivate.

As stated in the literature, for specific cell types, the proliferative potential may be inversely proportional to cell size [14,15]. Therefore, we evaluated cell size in each medium and at each of the first three passages (Figure 2). No clear correlation between doubling time (Figure 3) and cell size (Figure 2) was established, as the cells with the lower doubling time did not necessarily have the smallest size. Indeed, KCs showed no significant difference in cell size when cultured in both media or were even larger in SFM condition compared to FBSCM for Population 1 (Appendix A). At the same time, the doubling time was much more significant in FBSCM compared to SFM, especially in Passage 1. However, a greater doubling time was observed for the fibroblasts in SFM condition compared to FBSCM (see Figure 3). Fibroblasts are known to be quiescent cells in vivo, waiting to be activated through specific signaling. In the context of wound healing, they migrate, proliferate and synthesize extracellular matrix proteins into wound areas, favoring closure [19,20,21]. The standard cell culture conditions using growth factor-rich FBSCM induce an abnormal proliferation of fibroblasts which resembles non-physiologic conditions, with potential consequences, such as biases in the observed results. Therefore, if the objective of a SFM is to better reproduce normal in vivo signaling than FBSCM, an increased doubling time, as was observed in this study, may be expected. However, the medium could be improved/adapted to increase the proliferation of fibroblasts if the application requires it (e.g., for the production of skin substitutes by tissue engineering for patients with severe burns). In their study, Mariggió et al. showed that fibroblast proliferation could be enhanced by adding sodium hyaluronate (hyaluronan) and four amino acids (glycine, leucine, proline and lysine) [22]. In their study, Mast et al. showed that the addition of hyaluronan alone does not increase the proliferation of fibroblasts [23]. Afterwards, David-Raoudi et al. showed that the biological properties of hyaluronan depend on its molecular size [24]. Indeed, a moderate increase in fibroblast proliferation was observed using the native form of hyaluronan. In contrast, a sharp increase was observed using homemade hyaluronan with 12 and 880 saccharide units. Proline and glycine were already present in the SFM used in the current study. However, adding leucine and lysine could be an interesting way to improve fibroblast proliferation. On the other hand, SFM is equivalent to or better than FBSCM for epithelial cell expansion (including during extraction from biopsies and expansion to passage P0; see Appendix A). Due to its composition and impact on cell culture parameters, this SFM could be used for clinical applications (e.g., tissue engineering, cellular therapy).

Maintaining an adequate proliferative potential is essential in cell culture, primarily because it is known that the stem potential becomes exhausted over multiple passages. In the field of tissue engineering and in many other fields, epithelial cells are commonly used, studied or added to biomaterials to promote differentiation into a functional epithelium. However, to epithelialize biomaterials for long-term results, it is necessary to use a medium that does not deplete the proliferative potential of the cells, i.e., stem cells or progenitors, making it possible to maintain a healthy epithelium. This is important for clinical applications but also to allow the use of primary cell populations for numerous passages in fundamental research. In this study, a clonogenicity test has been performed to determine how many cells could still proliferate and form colonies after several passages. To evaluate the number of CFU, two techniques were used. First, a count of the cell colonies was done, but as this could be relatively subjective, we also scanned the stained flasks to evaluate the surface covered by rhodamine blue. Rhodamine blue specifically stains cells, allowing CFU area determination via the ImageJ software. The results of both techniques yielded the same results: at least the same quantity or a statistically more significant quantity of CFU was detectable in SFM compared to FBSCM, except for Keratinocyte Population 2 at week 3 (Appendix A). This result was expected, as the serum contained many components [3], some of which can promote cell differentiation rather than maintaining stem potential. In their study using adipose-derived mesenchymal stem cells (ASC), Devireddy et al. showed that cells cultured in a SFM presented more stem cell markers than FBSCM [25], which supports the idea of stemness potential preservation in SFM. However, Mattinger et al. showed that the maximum number of passages used in the cultivating human nasal epithelial cells varied from 2 to 11, depending on the commercial SFM used [26].

Conversely, Bhat et al. [27] cultured bone-morrow derived stem cells in several commercially available media and showed that none of the serum-free media tested achieved the multiple CFUs associated with the use of a medium-containing serum [27]. This discrepancy in the results presented in the literature can be explained by the specific nature of the signals inducing the differentiation process of stem cells according to their origin. The current SFM allowed epithelial (KC and UC, derived from ectodermal and endodermal lineages, respectively) proliferation maintenance in physiological conditions, which could be helpful in various applications. As hypoxia has been found to increase the maintenance of proliferative potential [28], using SFM in hypoxic conditions could be a means to further improve cell yields.

The culture medium impacts cell performance and behavior [27], which can cause cell metabolism modifications. Therefore, we assessed cellular glycolysis and mitochondrial respiration performance using a Seahorse XF extracellular flux analyzer. This technology allowed us to measure the extracellular acidification rate (ECAR) and the mitochondrial oxygen flux through the oxygen consumption rate (OCR) from living intact cells. Under most circumstances, the most important generator of cellular energy is mitochondrial respiration through mitochondrial oxidative phosphorylation [29]. On the other hand, glycolysis is another energy source which may or may not require oxygen, and is the first step of cellular respiration [30]. Nevertheless, glycolysis can also serve as an emergency backup for energy [30]. Except for the keratinocyte basal OCR and the VF maximal OCR, the basal and maximal OCR of all cell populations tended to increase in SFM compared to FBSCM (Table 1). These results show that the capacity to produce energy through mitochondrial respiration is equal to or increased in the cell culture using SFM compared to FBSCM. Therefore, the cellular metabolic behavior is closer to the physiologic one, with energetic metabolism coming mainly from mitochondrial respiration in the cell culture using SFM rather than glycolysis.

Furthermore, mitochondrial function is known to determine the bioenergetic cellular state, significantly contributing to cellular vulnerability or resilience [29]. Indeed, mitochondrial dysfunction is often associated with known pathologies [31,32,33]. For example, Pellerin et al. showed that cancer-associated fibroblasts (CAFs) exhibit modified metabolism, with increased basal and maximal glycolysis (ECAR) compared to normal fibroblasts [34] (also reviewed in [35,36,37]). The results obtained in the current study indicated that the ATP production through mitochondrial respiration and mitochondrial capacity tended to be greater in the cell culture conditions using SFM compared to FBSCM. From the perspective of studying cancer cells and their microenvironment, and as SFM provides a cell culture environment allowing cell metabolism to occur which closer to the native one than the environment provided by FBSCM, this medium could be a great cell culture medium candidate. Indeed, using a SFM with a known composition and in the absence of interfering growth factors and cytokines could make it possible to recreate an environment which is more likely to simulate cell–cell interactions than the FBSCM currently used in many studies.

The SFM described in this paper, in its current iteration, contained bovine albumin and insulin, i.e., animal-derived components which can easily be replaced by human recombinants. Although bovine albumin is a xenogenic protein, it is well-characterized, allowing us to include fatty acids in our lipid-supplemented medium. Human albumin could be used instead, but this would increase the cost [25]. We estimated that the SFM could be produced at a lower price compared to average FBSCMs for both fibroblast and epithelial cells. For now, small batches of SFM have been produced, and therefore, the price of each component remains high. However, if produced more significant quantities, savings could be made by purchasing the required components in bulk. This contrasts with the increasing cost of serums and the decrease in their availability, considering that only one-third of FBS produced met the regulatory requirements for cell therapy applications in 2003 [5,6].

The current SFM could be adapted for cell cultures intended for various clinical or fundamental applications. For example, a FBSCM has already been adapted for endothelial cell cultures [38]. The same conditioning could be applied to the current SFM to cultivate endothelial cells in serum-free and low-cost conditions. The latter could also be improved to cultivate certain cell types (e.g., neurons, adipose-derived stem/stromal cells), as cells are sensitive to their culture medium. Therefore, a range of products to cultivate cells in a much more physiological way could be developed, thereby overcoming the obstacles encountered when using serum in cell culture, e.g., biases in the results.

## 4. Materials and Methods

### 4.1. Ethics Statement

The current study was conducted according to the Declaration of Helsinki. It was approved by the committee for the protection of human participants (Comité d’éthique de la recherche du CHU de Québec-Université Laval, protocol code 2012-1341). All patients provided informed written consent prior to biopsies.

### 4.2. Cell Isolation and Culture

Urothelial cells (UC) and vesical fibroblasts (VF) were isolated from three human male bladder biopsies, whereas keratinocytes (KC) and dermal fibroblasts (DF) were isolated from three human female skin biopsies, as previously described [39,40,41]. Cell culture plastics were previously coated with Gelatin type A (Fisher Chemicals, 1% solution for 1 h) before using SFM. Fibroblasts (VF and DF) were thawed and used in Passages 3 or 4, whereas epithelial cells were used in Passages 0 or 1 for proliferative tests and Passages 1 or 2 for the other assays. Fibroblasts were seeded into culture flasks with a medium containing (or not) serum and incubated at 37 °C in a humidified 8% CO_2_ atmosphere. The medium was changed three times a week. Passages were performed when fibroblasts reached 80–90% confluence using trypsin (0.05% trypsin: ICN Biomedicals, Irvine, CA, USA; 0.01% EDTA: J.T. Baker, Phillipsburg, NJ, USA).

Epithelial cells were cultured in a FBSCM containing a 3:1 mix of Dulbecco–Vogt modification of Eagle’s (DMEM, Invitrogen, Burlington, ON, Canada) and Ham’s F12 (Flow Lab., Mississauga, ON, Canada) supplemented with 5% FBS (GE Healthcare, Chicago, IL, USA), 24.3 μg/mL adenine (Corning), 5 μg/mL crystallized bovine insulin (Sigma Aldrich, St. Louis, MO, USA), 1.1 μM hydrocortisone (Teva Canada Ltd., Toronto, ON, Canada), 0.212 μg/mL isoproterenol hydrochloride (Sandoz Canada, Boucherville, QC, Canada), 10 ng/mL epidermal growth factor (Austral Biologicals, San Ramon, CA, USA) and antibiotics, i.e., 100 U/mL penicillin and 25 mg/mL gentamicin (Sigma-Aldrich). Fibroblasts were cultured in DMEM supplemented with 10% FBS (Invitrogen) and antibiotics. A feeder layer of irradiated (6000 rad) human neonate foreskin DF was seeded at a density of 5333 × 10^3^/cm^2^ [42]. This seeding was done 14 days before epithelial cells were seeded. The medium was changed every seven days. Only half of the medium was changed when seeding the epithelial cells to keep it partly conditioned by the irradiated fibroblasts [43].

### 4.3. Constitution of the Serum-Free Medium (SFM)

The same cell populations applied to the medium-containing serum were used in the serum-free medium. Epithelial and fibroblasts cells were cultured in a medium that was partially inspired by studies using SFM [44] containing a 3:1 mix of Dulbecco–Vogt modification of Eagle’s (DMEM, Invitrogen, Burlington, ON, Canada) and Ham’s F12 (Flow Lab., Mississauga, ON, Canada) supplemented with 24.3 μg/mL adenine (Corning), 4 mM L-Glutamax (Sigma-Aldrich), 0.1 mM ethanolamine (Sigma-Aldrich), 0.1 mM O-phosphoryl-ethanolamine (Sigma-Aldrich), 20 pM triiodothyronine (Sigma-Aldrich), 0.2 µg/mL L-proline (Sigma-Aldrich) and 0.1 µg/mL glycine (Sigma-Aldrich).

To improve the medium performance, it was modified by the addition of 1X bovine serum albumin (Proliant, Feilding, New-Zealand), 1X lipid concentrate (ThermoFischer), 5.5 µg/mL transferrin (Sigma-Aldrich), 6.7 ng/mL selenium (Sigma-Aldrich), 10 μg/mL crystallized bovine insulin (Sigma Aldrich, St. Louis, MO, USA), 1.1 μM hydrocortisone (Teva Canada Ltd., Toronto, ON, Canada), 0.212 μg/mL isoproterenol hydrochloride (Sandoz Canada, Boucherville, QC, Canada), 10 ng/mL epidermal growth factor (Austral Biologicals, San Ramon, CA, USA) and antibiotics, i.e., 100 U/mL penicillin and 25 mg/mL gentamicin (Sigma-Aldrich) (UC medium). An overnight coating with gelatin (Sigma-Aldrich) was done before seeding cells in plates or flasks. The function of each component is explained in Table 3.

### 4.4. Proliferation and Size Determination

Fibroblasts and epithelial cells were thawed and seeded at 10% confluency in 12-well plates (5000 fibroblasts or epithelial cells/cm^2^ in 1 mL cell culture medium/well). The respective medium (FBSCM or SFM) was changed daily for four consecutive days. Each day, cells from three wells were harvested and counted separately using a Coulter-Beckmann Z2 (Beckman Coulter Life Sciences, Mississauga, ON, Canada) system which also provided the average cell size. A proliferation curve was performed to calculate the doubling time. The regression formula extrapolated from the proliferation curve was P = Po × e^gx^, where P is the number of cells on day x, Po is the cell number at day 0 and g is the exponential coefficient. Doubling time was calculated with the following formula: D = ln2/g (https://en.wikipedia.org/wiki/Doubling_time, accessed on 4 April 2022). On day 4 of Passages 1 and 2, cells were trypsinized and seeded for the subsequent passage.

### 4.5. Circularity Index Measurement

The cell cultures shown in 4.4 on day 4 of each passage (P1, P2 and P3) were photographed using a CKX41 light microscope (Olympus Corporation, Shinjuku-ku, Japan) with an E-620 camera (Olympus Corporation, Shinjuku-ku, Japan). After tracing the outline of the cells, the circularity index was determined using ImageJ software (NIH, Bethesda, MD, USA) with ten different cells for each image (*n* = 10).

### 4.6. Clonogenicity Determination

The protocol to determine the clonogenicity has already been described [16]. Briefly, irradiated human fibroblasts were first seeded for epithelial cells at a concentration of 1.3 × 105 in a 25 cm^2^ flask. One week later, 500 epithelial cells were seeded. The medium was changed once on day five. On day ten, the cells were fixed in 3.7% formol (ACP, Montreal, QC, Canada). Next, cells were stained with a Nile blue A/rhodamine mixture (Sigma-Aldrich) for 15 min and then rinsed three times with tap water, and the flasks were air dried at room temperature. Colony forming units (CFU) were then carefully counted directly in the flasks. Next, the flasks were scanned using a Typhoon trio + scanner (GE HealthCare, Cambridge Scientific, Watertown, NY, USA). ImageJ software was used to evaluate the stained area (NIH, Bethesda, MD, USA). Data are expressed as counted CFU/25 cm^2^ for the counts and stain/surface (AU) for the scans.

### 4.7. Metabolism Evaluation

The protocol to determine cellular metabolism has already been described [45,46]. Briefly, fibroblasts and epithelial cells (200,000 cells/cm^2^ in 100 μL cell culture medium/well) were seeded in XFe96 96-well plates. Before performing the analysis, cells were incubated for three days with the corresponding medium (FBSCM or SFM). Seahorse XFe96 sensor cartridge plates (Agilent/Seahorse Bioscience, Santa Clara, CA, USA) were hydrated the day before the analysis with the XF Calibrant (Agilent/Seahorse Bioscience, Santa Clara, CA, USA) and incubated overnight at 37 °C without CO_2_. Before the energy metabolism measurements, cells were washed and incubated for 1 h with Glyco Stress media or Mito Stress media. The Glyco Stress media contained XF Base Medium (minimal DMEM) (Agilent/Seahorse Bioscience, Santa Clara, CA, USA), supplemented with 2 mM L-glutamine (Wisent Bioproducts Inc., Saint-Jean-Baptiste, QC, Canada). The Mito Stress media contained XF Base Medium (Agilent/Seahorse Bioscience, Santa Clara, CA, USA), supplemented with 2 mM L-glutamine (Wisent Bioproducts Inc., Saint-Jean-Baptiste, QC, Canada), 1 mM sodium pyruvate (Wisent Bioproducts Inc., Saint-Jean-Baptiste, QC, Canada) and 10 mM D-(+)-glucose (Millipore Sigma, Oakville, ON, Canada). The extracellular acidification rate (ECAR), representative of glycolytic metabolism, and oxygen consumption rate (OCR), representative of mitochondrial respiration, were determined using an XFe Extracellular Flux Analyzer (Agilent/Seahorse Bioscience, Santa Clara, CA, USA). The concentrations indicated for each injection represented the final concentrations in the wells. The mitochondrial respiration was established by the sequential injection of 1.5 μM ATP synthase inhibitor oligomycin (67.5% oligomycin A complex) (Cayman Chemical, Ann Arbor, MI, USA), 0.5 μM of the mitochondrial uncoupler trifluoromethoxy carbonylcyanide phenylhydrazone (FCCP) (Cayman Chemical, Ann Arbor, MI, USA), and 0.5 μM of a combination of the mitochondrial complex I inhibitor rotenone (MP Biomedicals, Santa Ana, CA, USA) and mitochondrial complex III inhibitor antimycin A 0.5 μM (Millipore Sigma, Oakville, ON, Canada). The glycolytic metabolism was established by the sequential injection of 10 mM D-(+)-glucose (Millipore Sigma, Oakville, ON, Canada) and 1.5 μM of the ATP synthase inhibitor oligomycin (67.5% oligomycin A complex) (Cayman Chemical, Ann Arbor, MI, USA)—to inhibit mitochondrial respiration and force the cells to maximize their glycolytic capacity—and 50 mM 2-deoxy-D-glucose (2-DG) (Alfa Aesar, Ward Hill, MA, USA), a competitive inhibitor of glucose. At least three measurement cycles (3 min of mixing + 3 min of measuring) were completed before and after each injection. The oxygen consumption rate (OCR) and extracellular acidification rate (ECAR) were calculated using the Wave software v2.6 (Agilent/Seahorse Bioscience, Santa Clara, CA, USA). Energy metabolism was normalized according to the number of cells using a CyQuant Cell proliferation assay kit (Invitrogen, Burlington, ON, Canada), following the manufacturer’s instructions. The fluorescence of each well was measured at 485 nm/535 nm for 0.1 s using a Victor2 1420 MultiLabel Counter plate reader (Perkin Elmer Life Sciences, Waltham, MA, USA) and the Wallac 1420 software (PerkinElmer, Waltham, MA, USA). The normalized values were calculated from the fluorescence measurements with Microsoft Excel software (Microsoft, Redmond, WA, USA) and applied to the metabolic values.

### 4.8. Statistics

Statistical analyses were performed using the GraphPad Pri−sm v.9.2 software (San Diego, CA, USA). Results are expressed as means with standard deviation. In addition, a two-way Analysis of Variance (ANOVA) was used to interpret the data.

## 5. Conclusions

In this study, a SFM was compared to a cell culture medium containing fetal bovine serum (FBSCM), i.e., the most common type of serum used in the cell therapy industry [1]. Our results indicate that this SFM can enable fibroblast and epithelial cell proliferation while maintaining a microscopic aspect, cell size and metabolism roughly similar to those achieved with a FBSCM. The SFM was repeatedly found to been better suited for epithelial proliferation and clonogenicity. Fibroblasts and epithelial cells also showed greater mitochondrial respiration in the SFM compared to the FBSCM. However, the SFM could be adapted to further improve fibroblast proliferation. In conclusion, this SFM could be used for various applications, including better modeling of skin or urological cancers or to produce cells or tissue substitutes for clinical applications.

## Figures and Tables

**Figure 1 ijms-23-10035-f001:**
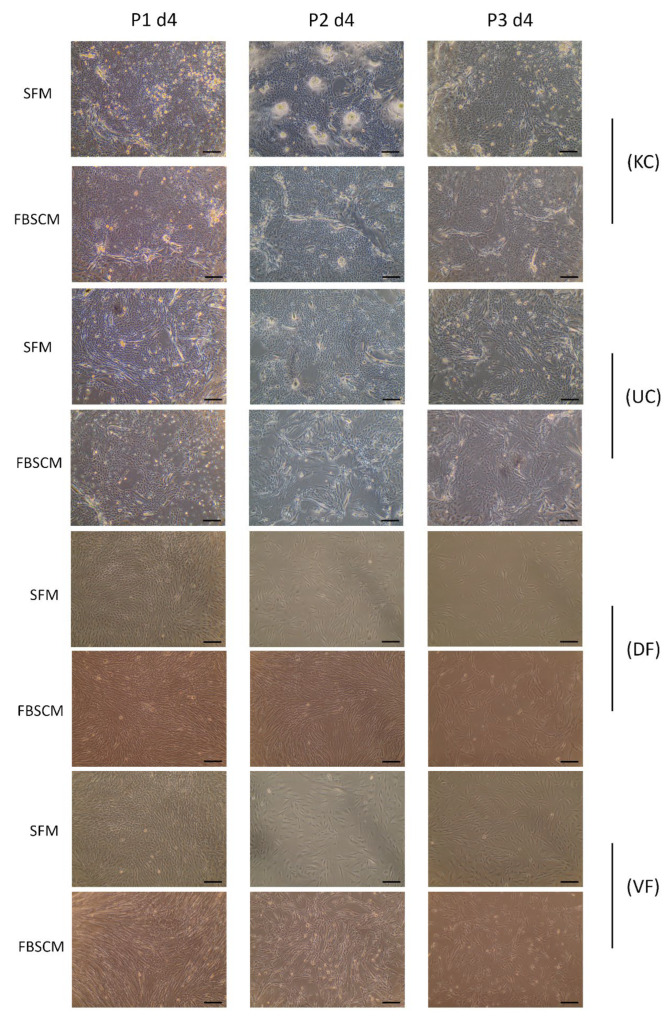
Morphology of epithelial and stromal cells at passages 1, 2 and 3. (KC) Morphology of keratinocytes, (UC) Morphology of urothelial cells, (DF) Morphology of dermal fibroblasts and (VF) Morphology of vesical fibroblasts after four days (P1 d4: passage one, day 4), 11 days (P2 d4: passage two, day 4) and 18 days (P3 d4: passage three, day 4) of culture in serum-free medium or fetal bovine serum-containing medium. Experiments were performed in triplicate for each cell population; a representative image is shown. The images were captured at 10× magnification. Scale bars = 100 µm.

**Figure 2 ijms-23-10035-f002:**
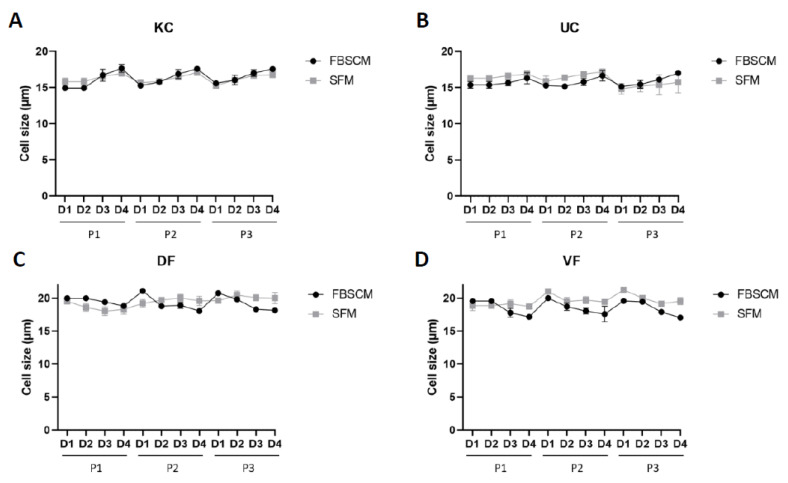
Cell size evaluation of keratinocytes (KC, panel (**A**)), urothelial cells (UC, panel (**B**)), dermal fibroblasts (DF, panel (**C**)) and vesical fibroblasts (VF, panel (**D**)) each day for a period of three passages. Each dot represents the mean of the cell size of three cellular populations, each realized in triplicate (N = 3, *n* = 3).

**Figure 3 ijms-23-10035-f003:**
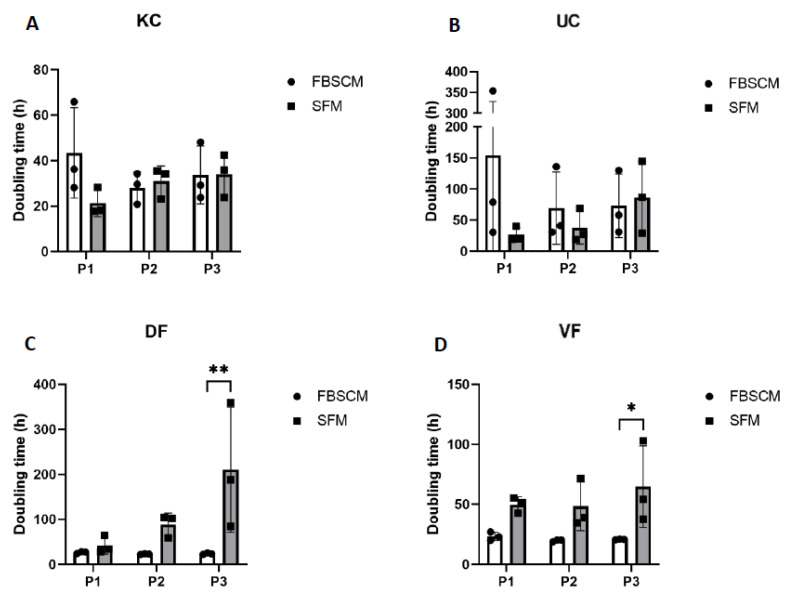
Doubling time of keratinocytes (KC, panel (**A**)), urothelial cells (UC, panel (**B**)), dermal fibroblasts (DF, panel (**C**)) and vesical fibroblasts (VF, panel (**D**)) over three passages. Ordinary two-way ANOVA was used to interpret the data. Asterisks indicate significant differences: (*) for *p*-value < 0.05, (**) for *p*-value < 0.01. Each dot represents the mean (N = 3) of a cell population cultured in triplicate (*n* = 3).

**Figure 4 ijms-23-10035-f004:**
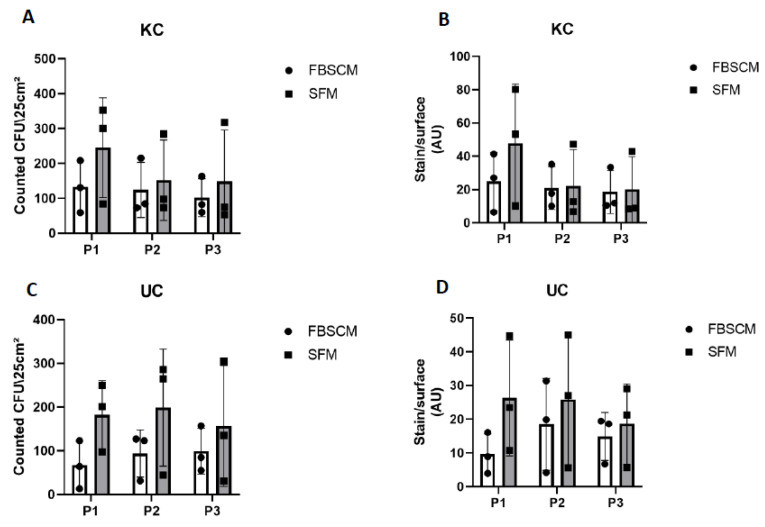
Clonogenicity evaluation of keratinocytes (KC) and urothelial cells (UC) over three passages. Clonogenicity was evaluated for KC by colony count (panel (**A**)) or stained surface measurement (panel (**B**)) and for UC by colony count (panel (**C**)) or stained surface measurement (panel (**D**)). Ordinary two-way ANOVA was used to interpret the data. Each dot represents the mean of a cellular population (N = 3) cultured in triplicate (*n* = 3).

**Figure 5 ijms-23-10035-f005:**
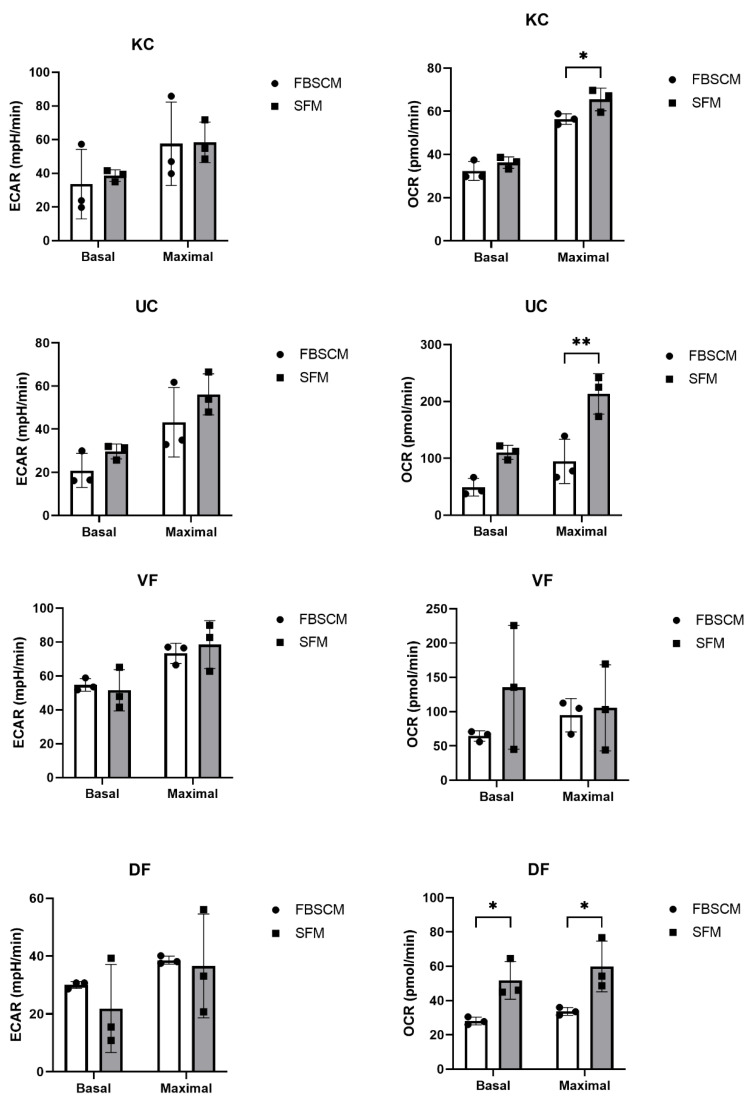
Extracellular acidification rate (ECAR) and oxygen consumption rate (OCR) evaluations of cell culture of keratinocytes (KC), urothelial cells (UC), dermal fibroblasts (DF) and vesical fibroblasts (VF). Ordinary two-way ANOVA was used to interpret the data. Asterisks indicate significant differences: (*) for *p*-value < 0.05, (**) for *p*-value < 0.01. Each experiment had five replicates for the three cell populations per cell type (N = 3, *n* = 5).

**Table 1 ijms-23-10035-t001:** Evaluation of the morphology of cells during their expansion.

	P1		P2		P3	
	FBSCM	SFM	FBSCM	SFM	FBSCM	SFM
KC	0.888 ± 0.064	0.860 ± 0.065	0.840 ± 0.037	0.855 ± 0.055	0.834 ± 0.062	0.870 ± 0.059
UC	0.838 ± 0.073	0.845 ± 0.065	0.814 ± 0.057	0.826 ± 0.071	0.832 ± 0.052	0.872 ± 0.065
DF	0.224 ± 0.049	0.283 ± 0.061	0.229 ± 0.062	0.243 ± 0.032	0.267 ± 0.069	0.249 ± 0.071
VF	0.235 ± 0.064	0.305 ± 0.066	0.230 ± 0.066	0.202 ± 0.040	0.239 ± 0.087	0.236 ± 0.054

The circularity indexes of epithelial and stromal cells at Passages 1, 2 and 3 were measured using the images presented in Figure 1. Ordinary two-way ANOVA was used to interpret the data. No significant difference was found. Each number represents the mean ± standard deviation (*n* = 10).

**Table 2 ijms-23-10035-t002:** Summary of the metabolism tendency results for all cell types cultured in SFM compared to FBSCM.

		KC	UC	DF	VF
ECAR	Basal	→	↗	↘	→
	Maximal	→	↗	→	→
OCR	Basal	→	↗	↗	↗
	Maximal	↗	↗	↗	→

**Table 3 ijms-23-10035-t003:** Cost evaluation of SFM and FBSCM according to their ingredients. Prices marked as <0.01 are inferior to 0.01 US$.

	Components	Price for a 500 mL Bottle (USD)
SFM	3:1 mix of Dulbecco–Vogt modification of Eagle’s and Ham’s F12	0.20
	L-Glutamax	6.42
	Epidermal Growth Factor	3.25
	Hydrocortisone	0.06
	Insulin-Transferin-Selenium (ITS)	34.22
	Lipid concentrate	4.56
	Triiodothyronine	<0.01
	Ethanolamine (M)	<0.01
	Isoproterenol	<0.01
	O-phosphoryl-ethanolamine (mM)	0.04
	Bovine Serum Albumin (BSA)	2.82
	L-proline	<0.01
	Glycline	<0.01
	Antibiotics:penicillin	<0.01
	Antibiotics:gentamycin	0.12
	TOTAL	51.50
FBSCM for stromal cells	Dulbecco–Vogt modification of Eagle’s medium	0.21
	Antibiotics:penicillin	<0.01
	Antibiotics:gentamycin	0.12
	FBS	75.60
	TOTAL	76.05
FBSCM for epithelial cells	3:1 mix of Dulbecco–Vogt modification of Eagle’s medium and Ham’s F12	0.20
	Epidermal Growth Factor	3.25
	Hydrocortisone	0.06
	Isoproterenol	<0.01
	Insulin	9.94
	Antibiotics:penicillin	<0.01
	Antibiotics:gentamycin	0.12
	Fetal Bovine Serum (FBS)	42.20
	TOTAL	55.77

## Data Availability

Data are available upon reasonable request.

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
