# Peer review of "Evaluation of a Serum-Free Medium for Human Epithelial and Stromal Cell Culture"

_ijms, 2022, doi:10.3390/ijms231710035_

Round 1
Reviewer 1 Report
This is an excellent article regarding the usage of a serum-free medium compared to the conventional cell culture with an FBS-containing medium. This study used four different cell types: Keratinocytes, urothelial cells, dermal fibroblasts and vesical fibroblasts. The main comparison is made wrt size distribution, doubling time analysis, clonogenic evaluation, and finally metabolic activity rates (EACR and OCR). However, In my opinion, we can not conclude the study unless either of the two more experiments is done;
1) Cell cycle analysis using flow cytometry.
2) STR authentication of the cells after different treatments.
The topic of this article is good. The paper is written carefully with nicely prepared figures. However, a major revision is required to improve the scientific content before the article is published.
Minor comment: Line 21; FBSCM should be Fetal Bovine Serum-Containing Medium.
Author Response
This is an excellent article regarding the usage of a serum-free medium compared to the conventional cell culture with an FBS-containing medium. This study used four different cell types: Keratinocytes, urothelial cells, dermal fibroblasts and vesical fibroblasts. The main comparison is made wrt size distribution, doubling time analysis, clonogenic evaluation, and finally metabolic activity rates (EACR and OCR). However, In my opinion, we can not conclude the study unless either of the two more experiments is done;
Thank you once again for the comments and suggestions.
1) Cell cycle analysis using flow cytometry.
In this study, we have focused on the expansion potential of the primary cell populations of mesenchymal or epithelial origins. We are interested in preserving the proliferative/metabolic potential of the mesenchymal and epithelial cells because these cells are commonly used in regenerative medicine. Our results show a slower proliferation of cells in SFM conditions vs FBSCM conditions for fibroblast populations, independently of the organ they originate from, but this was only statistically significant at passage 3. As mentioned in the discussion, fibroblasts are naturally slow-dividing cells, and we think the SFM recapitulates this characteristic. Nevertheless, a higher proliferation rate can be desired for fibroblasts to answer specific clinical needs, such as skin reconstruction for severely burned patients. Our culture conditions could further be modified. In this case, a comparative flow cytometry evaluation describing the proportion of cells in the different cell cycle phases according to media composition could be informative and serve as guidance in modifying the SFM. For the current study, we feel that the data provided is sufficient to document the impact of this SFM version of the media. So, we think cell cycle using flow cytometry would bring interesting supplementary results but would be outside the scope of the current study.
2) STR authentication of the cells after different treatments.
This study used only primary cell populations extracted from human biopsies (skin and bladder). The cells were cultivated for three passages (around nine doublings). We did not make specific clonal expansion or use established cell lines. Supplements used to reconstitute SFM are found in vivo and were used at nearly physiological concentrations. Moreover, we have not found, on PubMed, studies related to genetic instability or teratoma for these supplements. They are currently used in a clinical setting, especially in regenerative medicine. Considering this and in the context of this study, we have not performed STR verification. Nevertheless, in the future, we agree that investigating this will be particularly relevant if an application requires extensive cell amplification and long-term cell expansion.
The topic of this article is good. The paper is written carefully with nicely prepared figures. However, a major revision is required to improve the scientific content before the article is published.
Minor comment: Line 21; FBSCM should be Fetal Bovine Serum-Containing Medium.
Line 21 “Fetal Bovine Serum-Containing Serum (FBSCM)” was changed to "Fetal Bovine Serum-Containing Medium (FBSCM)."
Once again, we want to thank reviewer 1 for his/her helpful comments, which will help to strengthen our future studies on SFM refinement and applications.

Reviewer 2 Report
Remarks and comments for the authors of the article entitled: “Evaluation of A Serum-Free Medium for Human Epithelial and Stromal Cell Culture.“
The article entitled: “Evaluation of A Serum-Free Medium for Human Epithelial and Stromal Cell Culture” is useful and interesting especially for researchers carrying out experiments with cell cultures. The described results and discussion are promising in general, however, there are no genetic data results that could be shown the cell’s stability for tested cells cultured using media with- and without fetal bovine serum or some results showing the gene expression data for some selected genes for example related with energy metabolism to show more exactly some differences. Genetic research would significantly raise the level of this work.
Detailed remarks:
1) please standardize the description of the figures in the text with or without a space.
Author Response
Remarks and comments for the authors of the article entitled: “Evaluation of A Serum-Free Medium for Human Epithelial and Stromal Cell Culture.“
The article entitled: “Evaluation of A Serum-Free Medium for Human Epithelial and Stromal Cell Culture” is useful and interesting especially for researchers carrying out experiments with cell cultures. The described results and discussion are promising in general, however, there are no genetic data results that could be shown the cell’s stability for tested cells cultured using media with- and without fetal bovine serum or some results showing the gene expression data for some selected genes for example related with energy metabolism to show more exactly some differences. Genetic research would significantly raise the level of this work.
Thank you for the constructive comments.
As mentioned in the previous comment to reviewer #1, we have focused in this study on the expansion potential of the primary cell populations of mesenchymal and epithelial origins. We are interested in preserving the proliferative/metabolic potential of the mesenchymal and epithelial cells. Fibroblasts are naturally slow-dividing cells, and we think the SFM recapitulates this characteristic, reducing the risk of genomic instability. Furthermore, the cells were cultivated for only three passages (around nine doublings). Supplements used to reconstitute SFM are found in vivo and were used at nearly physiological concentrations.
Moreover, we have not found, on PubMed, studies related to genetic instability or teratoma for these supplements. However, they are currently used in a clinical setting, especially in regenerative medicine. Considering this and in the context of this study, we have not performed a genetic assessment. Nevertheless, in the future, we agree that investigating this will be particularly relevant if an application requires extensive cell amplification and long-term cell expansion. In this case, genetic stability according to media composition could be informative and serve as guidance to modify the SFM. Metabolomic and transcriptomic studies would indeed be helpful to characterize the cultures further when cell-specific SFM is established based on this initial study.
Detailed remarks:
1) please standardize the description of the figures in the text with or without a space.
The standardization has been done.
On the website, only one comment numbered 1) can be seen. We asked the Editor to be sure that the reviewer wrote no other comments. If there are unaddressed questions, we will be pleased to answer them.

Reviewer 3 Report
The manuscript entitled “Evaluation of a serum-free medium for human epithelial and stromal cell culture” by Caneparo et al. is a detailed account on some cellular and biochemical characteristics of different cell types cultured under serum-free or fetal bovine serum-containing medium. The manuscript comes with an extensive collection of regular and supplementary figures and tables.
Minor problems
1) I advise the authors to increase the size of the microscopic images on Figure 1 and Figures S1A-D because most of them are too small and have low resolution. The cells are varely recognizable; one should be able to assess the composition/purity and the fitness of a culture easily especially when these qualities are at the focus of this work.
2) Figure 3 shows KC and UC cells only, while its legend mentions DF and VF cells as well. Please clarify.
3) Lines 22-23: The authors claim that fibroblasts and epithelial cells cultured in SFM maintained “a morphology, cell size and metabolism similar to the FBSCM.” Contrary to this claim, however, cell morphology was not measured and analyzed. This should be corrected. (Some cell morphology parameters are: area, perimeter, lacunarity, density, span ratio, hull circularity, hull area, hull perimeter, max/min radii, mean radius, diameter of boundig circle, fractal dimension, roughness, transformation index, circularity (for details, see Fernández-Arjona et al., 2017).
Fernández-Arjona MDM, Grondona JM, Granados-Durán P, Fernández-Llebrez P, López-Ávalos MD. Microglia Morphological Categorization in a Rat Model of Neuroinflammation by Hierarchical Cluster and Principal Components Analysis. Front Cell Neurosci. 2017 Aug 8;11:235. doi: 10.3389/fncel.2017.00235. PMID: 28848398; PMCID: PMC5550745.
4) The number of cells for each cell types and for each passage should be given (line 408). This would complement data on “doubling time” as well.
5) Line 21: Containing Medium, not Serum
6) Line 210: >0.01 is correct?
7) Line 408: The “Coulter-Beckmann Z2 system” should be specified, city, country of the maker should be added.
8) Line 416: 4.5. How the clonogenicity of each passaged cultures was determined? The protocol should be detailed in the section “4.5. Clonogenicity determination”.
9) Lines 420-421: Staining of the cells should be detailed and indicated in figure legends (lines 420-421 are valid for all cultures?).
10) Line 422: City, country should be added for the scanner maker.
11) Line 423: The rationale and steps in section “4.6. Metabolism evaluation” should be referenced.
12) Lines 440-446: The choice of mitochondrial inhibitors should be referenced. Please indicate final concentrations for the inhibitors.
Author Response
The manuscript entitled “Evaluation of a serum-free medium for human epithelial and stromal cell culture” by Caneparo et al. is a detailed account on some cellular and biochemical characteristics of different cell types cultured under serum-free or fetal bovine serum-containing medium. The manuscript comes with an extensive collection of regular and supplementary figures and tables.
Minor problems
1) I advise the authors to increase the size of the microscopic images on Figure 1 and Figures S1A-D because most of them are too small and have low resolution. The cells are varely recognizable; one should be able to assess the composition/purity and the fitness of a culture easily especially when these qualities are at the focus of this work.
Thank you for your suggestion. We recognized that the pictures were small, making it difficult to observe differences. Unfortunately, we have to balance the size of our figures and the restriction size related to the publication. Therefore, we have increased the size of each supplementary data image by 70% and about 32% for figure 1. We hope the resized images allow better observation of the cells.
2) Figure 3 shows KC and UC cells only, while its legend mentions DF and VF cells as well. Please clarify.
Figure 3 (A to D) shows the doubling time of keratinocytes, urothelial cells, dermal fibroblasts and vesical fibroblasts. We have checked carefully using several computers without seeing the problem, but we will pay attention to ensure the whole figure is visible on all platforms.
3) Lines 22-23: The authors claim that fibroblasts and epithelial cells cultured in SFM maintained “a morphology, cell size and metabolism similar to the FBSCM.” Contrary to this claim, however, cell morphology was not measured and analyzed. This should be corrected. (Some cell morphology parameters are: area, perimeter, lacunarity, density, span ratio, hull circularity, hull area, hull perimeter, max/min radii, mean radius, diameter of boundig circle, fractal dimension, roughness, transformation index, circularity (for details, see Fernández-Arjona et al., 2017).
Fernández-Arjona MDM, Grondona JM, Granados-Durán P, Fernández-Llebrez P, López-Ávalos MD. Microglia Morphological Categorization in a Rat Model of Neuroinflammation by Hierarchical Cluster and Principal Components Analysis. Front Cell Neurosci. 2017 Aug 8;11:235. doi: 10.3389/fncel.2017.00235. PMID: 28848398; PMCID: PMC5550745.
We completely agree with the reviewer. Therefore, we added a new table summarizing the measurement of cells' circularity and added a supplementary figure to present these results in the form of a graph. We also added some text to introduce this new table and figure.
The morphology of three populations of keratinocytes (KC), urothelial cells (UC), dermal fibroblasts (DF) and vesical fibroblasts (VF) were evaluated microscopically (Fig. 1 and Table. 1). A representative image of each cell type has been selected for one cell population. The morphologies of the two other populations, which are from different donors, can be found in the supplementary data (supplementary Fig. S1). No significant difference in cell morphology was observed for epithelial cells or fibroblasts along the three passages in the two media.
Table 1. Evaluation of the morphology of the cells during their expansion.
|
P1 |
P2 |
P3 |
||||
|
FBSCM |
SFM |
FBSCM |
SFM |
FBSCM |
SFM |
|
|
KC |
0,888 ± 0,064 |
0,860 ± 0,065 |
0,840 ± 0,037 |
0,855 ± 0,055 |
0,834 ± 0,062 |
0,870 ± 0,059 |
|
UC |
0,838 ± 0,073 |
0,845 ± 0,065 |
0,814 ± 0,057 |
0,826 ± 0,071 |
0,832 ± 0,052 |
0,872 ± 0,065 |
|
DF |
0,224 ± 0,049 |
0,283 ± 0,061 |
0,229 ± 0,062 |
0,243 ± 0,032 |
0,267 ± 0,069 |
0,249 ± 0,071 |
|
VF |
0,235 ± 0,064 |
0,305 ± 0,066 |
0,230 ± 0,066 |
0,202 ± 0,040 |
0,239 ± 0,087 |
0,236 ± 0,054 |
The circularity index of epithelial and stromal cells at passages 1, 2 and 3 was measured on the images in Fig 1. Ordinary two-way ANOVA has been used to interpret the data. No significant difference has been found. Each number represents the mean ± standard deviation (n=10).
The materials and methods section was updated as follows:
4.5 Circularity index measurement. The cell cultures shown in 4.4, on day 4 of each passage (P1, P2 and P3), were photographed using a CKX41 light microscope (Olympus Corporation) with an E‑620 camera (Olympus Corporation). After tracing the outline of the cells, the circularity index was determined using ImageJ (NIH, Bethesda, USA) and done on ten different cells for each image (n=10).
An additional supplementary figure has been added as follows:
Figure S1E: Evaluation of the morphology of KC, UC, DF and VF at passages 1, 2 and 3.
Circularity index evaluation of keratinocytes (KC), urothelial cells (UC), dermal fibroblasts (DF) and vesical fibroblasts (VF) based on the images in Fig. 1. Ordinary two-way ANOVA has been used to interpret the data. No significant difference has been found for p-value<0.05. Each dot represents the circularity index of a cell within each cellular population, realized on ten cells (n=10).
4) The number of cells for each cell types and for each passage should be given (line 408). This would complement data on “doubling time” as well.
The number for each cell type seeded for the doubling time experiment establishment is now given at the lines 438-439 “Fibroblasts and epithelial cells were thawed and seeded at 10% confluency in 12-well plates (5 000 Fibroblasts/cm2, in 1 mL cell culture medium/well).” The text was therefore modified by “Fibroblasts and epithelial cells were thawed and seeded at 10% confluency in 12-well plates (5 000 fibroblasts or epithelial cells/cm2, in 1 mL cell culture medium/well).”
5) Line 21: Containing Medium, not Serum
We have done the modification.
6) Line 210: >0.01 is correct?
Thank you. It was a mistyping.
"Cost evaluation of SFM and FBSCM medium according to the different constituents. Prices indicated >0.01 are inferior to 0.01US$.” was replaced by “Cost evaluation of both SFM and FBSCM according to the different constituents. Prices indicated <0.01 are inferior to 0.01US$.”
7) Line 408: The “Coulter-Beckmann Z2 system” should be specified, city, country of the maker should be added.
The mentioned information has been added.
8) Line 416: 4.5. How the clonogenicity of each passaged cultures was determined? The protocol should be detailed in the section “4.5. Clonogenicity determination”.
9) Lines 420-421: Staining of the cells should be detailed and indicated in figure legends (lines 420-421 are valid for all cultures?).
The materials and methods section has been modified to include more explanation about the clonogenicity experiment.
The protocol to determine clonogenicity has already been described16. Briefly, irradiated human fibroblasts were first seeded for epithelial cells at a concentration of 1.3×105 in a 25 cm2 flask. One week later, 500 epithelial cells were seeded. The medium was changed once on day five. On day ten, the cells were fixed in 3.7% formol (ACP, Montreal, QC, Canada). Next, cells were stained with a Nile blue A—rhodamine mixture (Sigma-Aldrich) for 15 minutes and then rinsed three times with tap water, and the flasks were air dried at room temperature. Colony forming units (CFU) were then carefully counted directly on flasks. Then, flasks were scanned using a Typhoon trio + scanner (GE HealthCare, Cambridge Scientific, Watertown, USA). ImageJ software was used to evaluate the stained area (NIH, Bethesda, USA). Data are expressed as counted CFU/25cm2 for the counts and stain/surface (AU) for the scans.
10) Line 422: City, country should be added for the scanner maker.
The mentioned information has been added.
11) Line 423: The rationale and steps in section “4.6. Metabolism evaluation” should be referenced.
12) Lines 440-446: The choice of mitochondrial inhibitors should be referenced. Please indicate final concentrations for the inhibitors.
Two references have been added.
This sentence has been added: The concentrations indicated for each injection represent the final concentrations in the wells.

Round 2
Reviewer 1 Report
Considering the review reply, I think the article can be published in the present form. Thanks to the authors for the review reply!